# Spatial Interaction Analysis of Shared Bicycles Mobility Regularity and Determinants: A Case Study of Six Main Districts, Beijing

**Lujin Hu** , **Zheng Wen, Jian Wang \* and Jing Hu**

School of Geomatics and Urban Spatial Informatics, Beijing University of Civil Engineering and Architecture, Beijing 102616, China
* Correspondence: wangjian@bucea.edu.cn

**Abstract:** Understanding the regularity and determinants of mobility is indispensable for the reasonable deployment of shared bicycles and urban planning. A spatial interaction network covering streets in Beijing's six main districts, using bike sharing data, is constructed and analyzed. as Additionally, the exponential random graph model (ERGM) is used to interpret the influencing factors of the network structure and the mobility regularity. The characteristics of the spatial interaction network structure and temporal characteristics between weekdays and weekends show the following: the network structure on weekdays is obvious; the flow edge is always between adjacent blocks; the traffic flow frequently changes and clusters; the network structure on weekends is more complex, showing scattering and seldom changing; and there is a stronger interaction between blocks. Additionally, the predicted result of the ERGM shows that the influencing factors selected in this paper are positively correlated with the spatial interaction network. Among them, the three most important determinants are building density, housing prices and the number of residential areas. Additionally, the determinant of financial services shows greater effects on weekdays than weekends.

**Keywords:** spatial interaction network; mobility regularity; shared bicycles; six main districts in Beijing; exponential random graph model (ERGM); determinants

## 1. Introduction

Since 2014, the use of shared bicycles has gradually increased, becoming a new type of "Internet + sharing" travel pattern. Especially after the COVID-19 pandemic, the bike sharing system (BSS) has gradually become one of the most widespread and popular shared and sustainable mobility systems [1–3], providing a viable alternative mode of transport for short or medium urban distances [4]. The BSS is generally equipped with positioning devices to record location in real time, contributing much data for urban flow analysis in cities [5,6]. The current bike sharing system still cannot successfully solve the problem of rebalancing [7–11]. During peak periods, areas with a high demand for shared bicycles tend to suffer from low supply [12]. In areas with low demand, shared bicycles constitute a large amount of urban resources that cannot be used. The key to the success of shared bicycles lies in whether they can meet the needs of users, and the driving factors of these needs [13]. Since the establishment of the BSS around the world, in order to improve the BSS and solve the demand problem, many studies had focused on travel behavior, the regularity of shared bicycle use in cities, the spatiotemporal pattern, travel demand and influencing factors [14–16]. Therefore, this paper focuses on the travel rules and influencing factors of shared bicycles.

Research on the influencing factors of population migration between cities or regions is important in the fields of urban economics, regional economics and geographical economics [17–20]. Although the use of shared bicycles is also a form of urban travel behavior, their influencing factors are quite different to those of population migration and other such ways. At present, much research has focused on the influencing factors of shared

bicycle travel [21–23], and showed that the travel pattern of the shared bicycles has obvious spatiotemporal characteristics and different determinants; a variety of factors, including weather, built environment, land use, public transport, station levels, socio-demographic factors, time factors, and safety, all play a role [14]. Among them, convenience is the most important factor [24,25], population density [26,27], flow conditions, temporal characteristics and weather [28] are factors also having an impact. Different areas such as rail transit stations, railway-stations, and the built environment around the stations are also the main influencing factors [29]. Factors such as campus characteristics, time factors, and weather events have a positive impact on campus bike-sharing travel [28]. Convenience facilities are found to have a positive impact at public places such as public transportation centers, entertainment centers, and coffee shops [30]. The previous studies show that the influencing factors of shared bicycles are indispensable for improving the BBS and guiding the reasonable layout of bicycles. At present, the research on the influencing factors of shared bicycles mostly adopts regression analysis and correlation analysis methods [22–30]. Although these methods can qualitatively and quantitatively analyze the effects of various factors, it is difficult to study their effects on the interaction between regions by taking the properties of regions as factors.

In addition, the city flow can be divided into daily flow (weekday flow), rest day flow and other flow [31]. Due to the large difference in travel purposes between weekdays and weekends, the traffic attributes of these two periods are completely different, and the spatial agglomeration characteristics of residents and the population's center of gravity are also different [32,33]. Therefore, the differences in the travel regularities of shared bikes and the influencing factors of different periods should also be paid attention to [34], i.e., the travel regularity of shared bikes at weekdays and weekends (holidays). The model without consideration of different periods is not so applicable for accurate spatiotemporal analysis. Researchers found obvious differences in shared bicycles' travel patterns, time concentration [21] and travel distance [35] on weekdays and weekends. Seasonal differences in flow profiles were identified for each day of the week [36]. Therefore, a comparative study of travel behavior based on weekdays and weekends (holidays) can reveal the characteristics of travel behavior at different periods, in order to explore residents' travel regularity. However, the current research on the differences in the utilization of shared bicycle between weekdays and weekends mostly focuses on the regularities of the travel flow [34,36–38], while a few of them described and analyzed the spatiotemporal variation of the influencing factors. By analyzing the impact of different factors on shared bicycles at different times, we can better understand and improve BSS from a new perspective. In order to improve the shortcomings of previous research, we innovatively construct a complex network model based on shared bicycle trip data and use the network statistical model. From the perspective of the network, we conduct an overall analysis of the shared bicycle system in the city, and qualitatively and quantitatively analyze the travel patterns and influencing factors of shared bicycles in different periods within the same week.

Since the 1970s, network science has undergone rapid development [39]. With the applications of complex network theory in many fields such as biology [40] and social science [41], urban and regional networks have also attracted widespread attention [42–44]. Public travel activities among regions can be used to construct a typical directed weighted spatial network, so that the travel behavior in different regions can be revealed from the constructed spatial network. There are many methods of analysis of the spatial interaction network, including characteristics and statistics analysis of the network, directed weighted network statistical analysis [45], network centrality theory [46], community detection theory [18,47] and network prediction analysis. The vertexes relationship of a network can be predicted and analyzed from the constructed statistical network model and its attributes. According to the complexity of the network model, the statistical network model can be divided into the simple random graph model [48], dyadic independence model [49], dyadic dependence model [50] and high-order dependence model [51]. The differences between observation networks and randomly occurring networks can be explained by the network

statistics and modeling. The weights of network member attributes and global characteristics are used to determine the degree of influence of each factor. Complex network theory enriches the methodology of research on shared bicycle travel behavior. Hence, network theory is introduced into the study of the travel regularities of shared bicycles in cities [52,53]. By studying the travel network, geographical divisions of shared bicycles can be found between areas with different degrees of demand [52], with a correlation between the flow strength and point-of-interest (POI) [53]. The research areas of the city are divided into different areas according to the required spatial scale, and areas are regarded as nodes of the network graph. Different from previous research methods, the complex network structure is constructed by travel activities such as shared bicycles between areas. The network regards the region as an independent individual in the system, and the research on travel laws and influencing factors is based on this independent unit of the region. Therefore, the method can comprehensively consider and analyze the internal factors of the region, which helps optimization from the perspective of the entire network, and directly serves the development and construction of the region.

As one of the biggest cities in the world, Beijing has a huge population. As of September 2021, the total number of shared bicycles in Beijing reached 941,000 [54]. In Beijing, the six administrative districts of Xicheng, Dongcheng, Chaoyang, Haidian, Fengtai and Shijingshan are collectively referred to as the Urban Six Districts. Approximately 60% of Beijing's total population and nearly 70% of its industries are located in the Urban Six Districts of Beijing, and it is the capital's core functional area. The Urban Six Districts have a vast area and a huge population, which makes the flow of people complex, with the characteristics of long range, long time and multiple methods of transport. Particularly during the peak hours of weekdays, serious road congestion causes shared bicycle travel to have a completely different spatiotemporal pattern and regularity compared with other countries and regions. Coupled with the impact of COVID-19 on people's travel patterns, it is urgent to study the travel regularity of shared bicycles and their influencing factors. Not only will such studies reveal important practical values of the rational layout of shared bicycles, and good related policies, but they will also provide an important reference for the construction of BSS in other cities. With the process of urbanization, administrative division is always studied at the spatial scale of street blocks. Hence, street blocks are used as the basic research units, which can directly provide suggestions for the decision making and planning of the street-related management departments. All instances of "streets" in this paper refer to the administrative division unit of street blocks.

For this reason, three problems are proposed and resolved in this paper. How can we explore the spatial characteristics and geometric structure of shared bicycles traveling between streets from the perspective of the network model? What are the influencing factors of the inter-street shared bicycle travel network in the Urban Six Districts of Beijing? What are the differences between the travel patterns of shared bicycles on weekdays and weekends? To solve these problems, this paper attempts to (i) analyze the structure of the shared bicycles' network and analyze its regularity; (ii) construct the exponential random graph model (ERGM), combining pure network structure variables and node attribute variables, and explore the core influencing factors affecting the formation of shared bicycle travel networks within the Urban Six Districts of Beijing; and (iii) analyze the spatiotemporal characteristics and influencing factors of shared bicycle travel behavior during weekdays and weekends.

This paper is organized as follows: the research area and data sources, the principles of network construction, and the indicators for the ERGM model are introduced in the second part, and the network structure and indicators are analyzed in the third part. The conclusions and a discussion are given in the fourth part.

## 2. Materials and Methods

*2.1. Research Data*

Six districts of Beijing (Dongcheng, Xicheng, Haidian, Fengtai, Shijingshan, and Chaoyang) were chosen as the basic research areas. Due to the limitation of the operating area of shared bicycles, the research area chosen in this paper was the area in which the six districts of Beijing intersect with the operating area of shared bicycles. A total of 122 street-level administrative units of this area were selected as the research units (in Figure 1b the specified intersected research area is extracted). Two types of data sources were used (see Table 1):

(1) Shared bicycle data: The shared bicycle data used in this paper is enterprise cooperation data, with the attributes of bicycle ID number, time, and the longitude and latitude of the bicycle. The data collection time interval was 3 h, and these two points of collected data were correlated with the ID number of the bicycle. The starting and ending points of each bicycle were obtained to generate a trajectory. Taking into account the impact of the new crown pneumonia epidemic, the data collection period was from early May to mid-to-late August. During this period, the epidemic situation in various regions of China, especially Beijing, was relatively optimistic, and the impact on travel was small, with residents being more willing to use shared bicycles to travel. Screening the existing data: First, in the existing data, abnormal data (such as rain, high temperature, etc.) were removed. Then, the pre-experiment of the data was carried out. The experimental results show that the distribution of shared bicycles on each weekday or weekend in the same week is basically stable, and the distribution of different weeks shows seasonal fluctuations on a weekly basis. Finally, after excluding the data of abnormal day (such as the day with rain, high temperature, etc.), Among the remaining data of 17 May 2021 to 23 May 2021 is a week with continuous date series and continuous data records, and it conforms to the general rule of bicycle distribution; 17 May to 21 May are weekdays, and 22 May and 23 May are weekends. In order to compare the difference between weekdays and weekends, the working days were divided into three parts for a better analysis of the travel regularities: the period of 17 May, the period of 18 May and the period of 19 to 21 May. We divide the data of weekdays in this way, on the one hand, to make the data of weekdays and weekends continuous, on the other hand, Monday is often the busiest day of the week in China, the regularity of the day is important for the travel pattern analysis of a week. Additionally, the number of data collection intervals on 17 and 18 May are the same, and the number of data collection intervals on 19 and 21 May and 22 to 23 are the same. Because 17:00 and 20:00 are the start and end time of the evening rush hour on weekdays, the data of this period of a day are chosen to show the spatial distribution characteristics of the shared bikes. Meanwhile in order to compare the differences of the distribution of shared bicycles on weekdays and weekends, we chose one day on weekdays and weekends to display the distribution rules of the shared bicycles. The results of the kernel density analysis of shared bicycles' spatial distribution are shown in the figure below (Figure 2).

(2) Other data: With the rapid development of electronic maps and location-based services, points of interests (POIs) have been continuously enriched and improved. POIs have the attributes of spatial information, such as latitude, longitude, and address, and other information, such as names and category. POIs can reflect the distribution of various facilities and institutions and are used in this paper, including attributes such as transportation facilities, financial services, residential areas, shopping, catering, and education institution (Figure 3a–e). The main types of POI spatial distribution are shown in the figure below. Besides POI data, road net-work data (including urban expressways, subways, expressways, national roads, provincial roads, county roads, and nine-level roads, etc.), building data and the housing prices of each street block are used as indicators of the travel patterns of shared bicycles in

this paper (Figure 3f). The data of the above content can also be downloaded from https://master.apis.dev.openstreetmap.org (accessed on 26 April 2021).

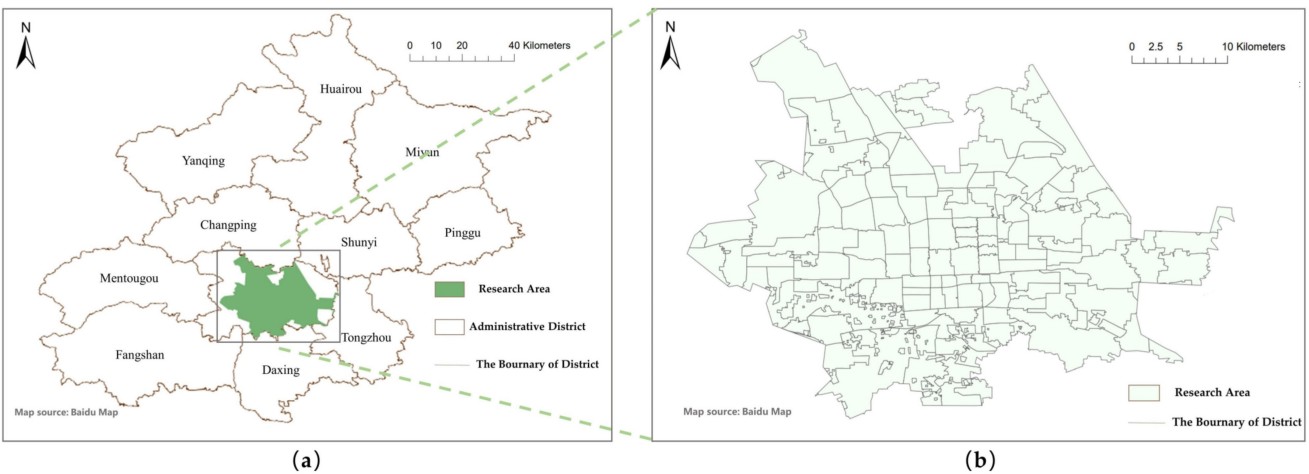

**Figure 1.** (**a**) Research area in Beijing; (**b**) Research Area of the darkened place in (**a**).

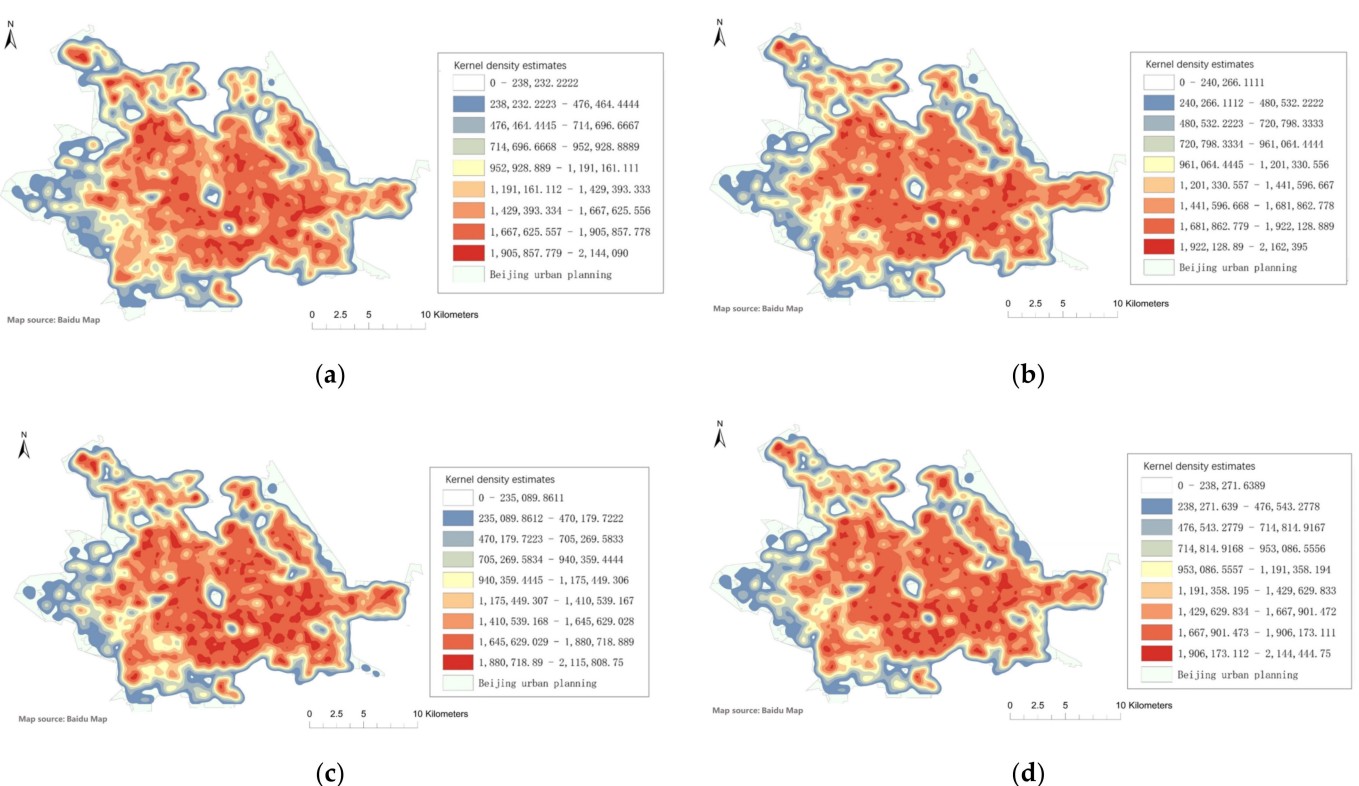

**Figure 2.** Heat map of shared bicycle spatial distribution at 17:00 and 20:00 from 19 May 2021 to 22 May 2021: (**a**) 17:00 19 May 2021; (**b**) 20:00 19 May 2021; (**c**) 17:00 22 May 2021; (**d**) 20:00 22 May 2021.

### 2.2. Research Methods

#### 2.2.1. Travel Network Construction

The network structure is composed of a series of actors and their relationships. The actors are called vertices or nodes, and the relationships between actors are called edges, or connections. This relationship can be direct or indirect. All pairs of nodes are stored in the form of two-tuples. In a directed network, two-tuples can establish relationships through asymmetric or mutual interactions.

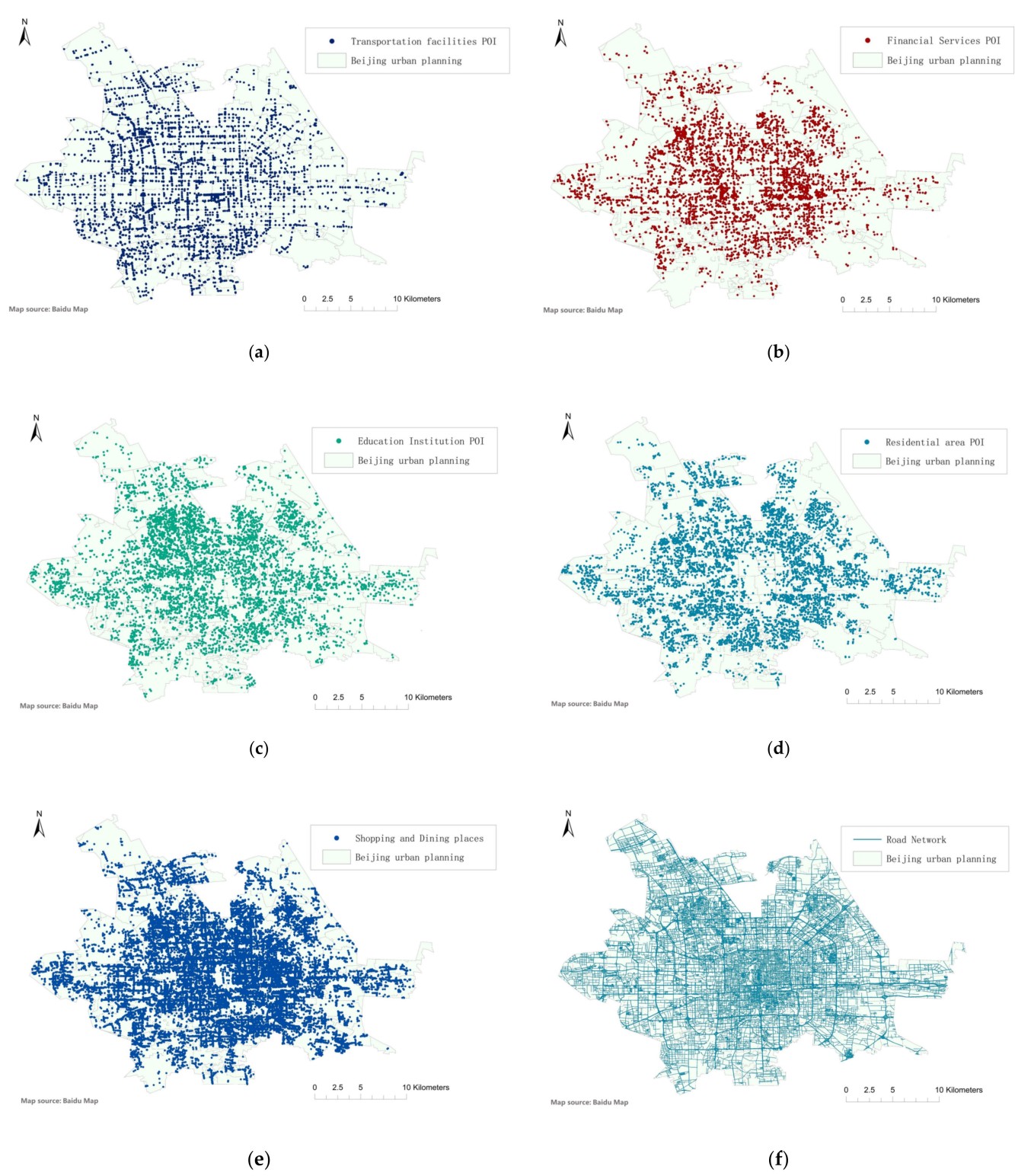

**Figure 3.** Spatial distribution of data in the research area: (**a**) POI of transportation facilities; (**b**) POI of financial services; (**c**) POI of education institution; (**d**) POI of residential area; (**e**) POI of shopping and dining places; (**f**) Road network in Urban Six Districts of Beijing.

According to the principle of network construction, residents' travel activities, considering streets as nodes, will form a corresponding directed network. The travel behavior is represented by a directed edge between nodes, in which the direction of the edge is the direction of the travel activities, and the weight of the edge is equal to the number of travel

activities. Therefore, the location points of the same bike acquired at different times can be associated with the ID number of the shared bike, and the starting and ending nodes can be determined by the location point within the street block. Then, a travel trajectory between streets can be generated. With the directed edges between nodes, a network structure with streets as nodes is finally constructed (Figure 4).

**Table 1.** Data sources.

| Type of Data | Number of Data Pieces | Data Source | Update Date |
| --- | --- | --- | --- |
| Shared bicycle data | 1,903,655 | Hello Bike | May, 2021 |
| POIs data | 168,271 | Baidu | April, 2021 |
| Road network data | 103,592 | Baidu | April, 2021 |
| Building data | 283,088 | Baidu | April, 2021 |
| Housing prices data | 122 | Baidu | April, 2021 |

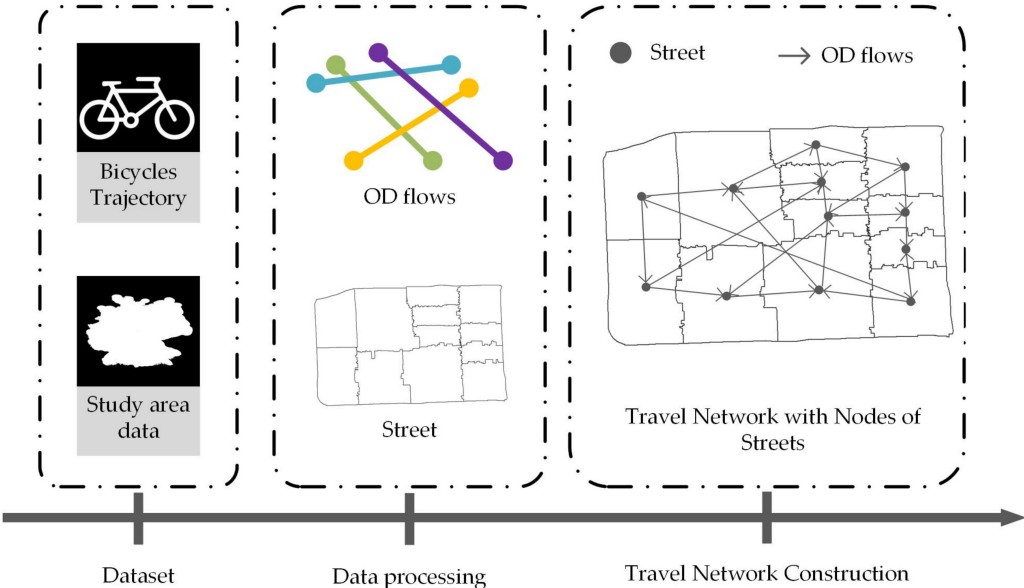

**Figure 4.** Travel network construction. The data used in this paper include shared bike trajectories and study areas. The OD flows are generated from the shared bicycle trajectory, and the streets in the study area are extracted as network nodes. The location of the start and end points of the OD stream are determined by the location of the block. A travel network is generated with streets as nodes.

2.2.2. Characterization of Travel Network

Network statistics can reflect the crucial information of a network's characteristics. Several standard network statistics metrics can be computed to characterize a weighted directed network [55], as presented below:

Network density: This is an index used to describe the degree of network relevance. This indicator can characterize the number and complexity of relationships in the network, and is a measurement of the completeness of the network structure. The calculation equation is shown in Equation (1).

$$D = \frac{d}{k(k-1)} \tag{1}$$

where $D$ is the network density, $k$ is the number of nodes, and $d$ is the number of edges in the network. When $D = 1$, it means that all nodes in the network are connected, and when $D = 0$, it means that there is no connection between nodes. The larger the value of the network density, the more complex the network structure, the closer the relationship between network nodes, and the larger the degree of mutual influence between nodes.

In the undirected network, the strength of a node is defined as the sum of the weights of all edges connected to the same node:

$$S_i = \sum_{j \in N_i} W_{ij} \tag{2}$$

$N_i$ is the set of nodes connected to node $i$, and $W_{ij}$ is the weight of the edge connecting nodes $i$ and $j$, which in this paper is the number of shared bicycles. The network structure of this paper is a directed network, so according to the direction of the edge, the node strength can be divided into the in strength and the out strength of the node [56]. According to the equation calculation, the net flow ratio (NFR) of the node can be obtained [57]. The calculation equation is shown in Equations (3)–(5).

$$S_{in}(v_i) = \sum_{v_j \in V_{in}} w(v_j, v_i) \tag{3}$$

$$S_{out}(v_i) = \sum_{v_j \in V_{out}} w(v_j, v_i) \tag{4}$$

$$NFR = \frac{S_{in}(v_i) - S_{out}(v_i)}{S_{in}(v_i) + S_{out}(v_i)} \tag{5}$$

$S_{in}(v_i)$ and $S_{out}(v_i)$ are the incoming and outgoing intensity, respectively, $V_{in}(v_i)$ is a set of nodes flowing to node $i$, $V_{out}(v_i)$ is the set of nodes flowing from node $i$ to other nodes, and NFR is the net flow ratio. When NFR > 0, it means that the incoming intensity of the node is larger than the outgoing intensity, and the popularity of the place is high. Otherwise, the popularity of the place is low.

### 2.2.3. Analysis of the Determinants of the Travel Network

The Exponential Random Graph Model (ERGM) is a relational data measurement model, which has been applied to the analysis of social networks both home and abroad. It is a newly developed and popularized model [45]. Different from general descriptive methods, ERGM is a statistical model, which can not only explain the observed characteristics of the network structure, but also explain the internal mechanism of network formation [39].

A random graph $G = (V, E)$ represents the random spatial correlation matrix associated with $G$, and $M_{i,j}$ represents a directed edge from $i$ to $j$ in $G$, and $M = [M_{i,j}]$ represents the adjacency matrix. The elements in the $i$-th row and $j$-th column correspond one-to-one with the relationship from node $i$ to node $j$. $P_\theta(M = m)$ represents the probability of $m$ appearing in $M$ under the condition of $\theta$, assuming that the probability of network formation depends on the various network structure statistics included in the model. The general and simplest form of ERGM is shown in Equation (6).

$$P_\theta(M = m) = \left(\frac{1}{k}\right) exp\left\{\sum_H \theta_H g_H(m)\right\} \tag{6}$$

$$k = \sum_m exp\left\{\sum_H \theta_H g_H(m)\right\} \tag{7}$$

In Equations (6) and (7), $g(m)$ represents the network statistics related to matrix $m$, which mainly includes pure network structure statistics that may affect the formation and construction of network relationships, network structure statistics related to node attributes, and network structure statistics related to other external relationship networks. $\theta$ is the estimated parameter vector of the relevant network statistics. If the parameter is positive (negative), it indicates the probability of the statistic appearing being more (less) accurate than the random expectation under control of other statistics; this statistic has a positive (negative) influence on the formation of the network structure. $H$ represents a series of structural statistical variables for the network. The magnitude of this value can also reflect the influence of the statistic on the formation of the network. $k$ is the normalization constant

of the above-mentioned network statistics, which can cause the sum of the estimated parameters of all network structure statistics to be 1. The calculation equation is shown in Equation (7).

### 2.2.4. Network Structure Variables

From the perspective of network formation, any network structure is a special case among many network structures formed by various nodes in the network. ERGM aims to describe the connection or structure in this network, and explain the reasons for the formation of the network structure and connection relationship. The explanatory variables (influencing factors) are expressed in the form of probability in the process of forming the network. Therefore, the explanatory variables are actually a series of network structure statistics.

The network structure statistics enable a reasonable explanation for the formation of the ERGM network structure. The number of network edges, mutuality, transitivity, interactive k star, interactive k path, etc., are the common statistical methods used for the ERGM. Studies have shown that the network should have a strong convergence, while if triangular structure variables are added to the ERGM model estimation, this may cause the attenuation of the model and the non-convergence of the estimated parameters [58]. It has been verified that the network convergence is determined by the number of edges and reciprocity added into the model. The statistics and significance of each variable are shown in Table 2. In addition to the statistics inside the network, other statistic indicators can explain the network structure, which are the attribute variables of the nodes. The probability of forming an edge is affected by the attributes of the two linked nodes. ERGM can input individual attributes into the model by adding statistics to the node. The influencing factors selected in this paper are input as the node attribute variables. In the model, the selected factors are described in detail, as follows:

(1) Housing Prices (Price): Housing is an important part of the national economy, and housing prices directly affect people's housing choices. The relationship between housing prices and travel behavior is inseparable, and there is obvious spatial heterogeneity between them [59]. We calculate the average housing prices of streets as one of the influencing factors;

(2) Building Density (Bden): Residents' travel in the city is affected by the distribution of buildings [56], which can be directly expressed as building coverage or building density. Building density is an important indicator used to measure urban development, which can reflect the density and vacancy rate of street blocks and buildings. Therefore, building density was chosen as one of the influencing factors;

(3) Road Network Density (Rden): Similar to building density, road network density is also one of the standards used to measure the development of a city. It can evaluate the rationality of a city's road network and represent the basic road network development. Researchers found that the density of the road network has a great impact on the daily life of residents, especially in terms of bicycle travel [57,60];

(4) Transportation Facilities (Tra): Traffic accessibility has a positive impact on the number of shared bicycle trips [61,62]. Traffic accessibility is affected by the distribution of traffic facilities, and affects residents' travel. Transportation facilities mainly include transportation hubs such as bus stations and subway stations. For POI data containing the attributes of transportation facilities, it is used to represent the distribution of transportation facilities. The bus stations and subway stations in the POI data are mixed with other types of transportation facilities, and then, the number of POI which contain the attributes of transportation facilities is calculated in each street as the input of the model;

(5) Shopping and Dining Places (SaD): The number of cafes and restaurants has a significant impact on the use of shared bicycles [30]. In addition, the distribution of shopping institutions such as malls and supermarkets has an impact on the spatial agglomeration of crowds on weekends and holidays [63]. Therefore, the POI of shop-

ping and dining attributes are mixed together to reflect the convenience of shopping and dining in each street block, which is treated as one of the influencing factors;

(6) Financial Services (Fin): Financial service facilities have a significant impact on shared bicycle travel [64]. The financial service industry has experienced the most significant productivity growth in the modern urban service industry. It is one of the industries that use modern information technology with a higher density, which has attracted the widespread attention of economic geographers [65]. In the previous studies on the factors affecting shared bicycle travel patterns, financial service factors are rarely considered. Hence, POI data with financial service attributes are chosen as the standard to measure the financial service capabilities of the street block;

(7) Residential Area (Res): Different from building density, residential density directly reflects the residential attributes in the area. In China, there are four indicators of residential density: residential population density, residential land per capital, residential building density and residential building area density. Studies have found that the residential density and POI of residential attributes have a certain impact on the travel demand of shared bicycles [56,66]. Therefore, the POI data of residential area attributes are chosen as one of the influencing factors;

(8) Education Institution (Edu): On weekdays, science and education services have a positive impact on shared bicycle travel [22]. There are several scientific education institutions for every street block, which can effectively improve the convenience of people's lives. According to the microscopic neoclassical theory, maximizing personal interests is the main reason for crowd mobility. Therefore, POI data with the attributes of science education are used to measure the convenience of education services.

**Table 2.** ERGM variables description.

| Variable | Schematic Diagram | Statistics | Statistical Significance |
|---|---|---|---|
| Edges | ○ ⟶ ○ | $\sum_{ij} y_{i,j}$ | Similar to linear regression constant term, generally not explained |
| Mutual | ○ ⟷ ○ | $\sum_{ij} y_{i,j} y_{j,i}$ | Tests the mutual relationship between different street blocks |

In summary, the influencing factors of the network structure statistics are used in this paper, and are shown in Table 3.

**Table 3.** Definitions of variables in the ERGM model.

| Variable | | Measure Unit | Definition |
|---|---|---|---|
| Network structure effects | Edges | Number | Equal to the number of links in the network |
| | Mutual | | Whether network nodes tend to form interactions |
| Node attribute effects | Price | Thousand CNY per square kilometer | Average housing prices in the neighborhood |
| | Bden | % | Street building density |
| | Rden | Kilometer per square Kilometer | Street road network density |
| | Tra | Number | Number of flow POI in the street, such as buses and subways |
| | SaD | Number | Number of shopping and dining POI in the street |
| | Fin | Number | Number of financial service POI in the street |
| | Res | Number | Number of POI in residential areas in the street |
| | Edu | Number | Number of education institution POI in the street |

## 3. Results

*3.1. Network Structure Analysis*

According to the principle of network construction, the shared bicycle travel network is constructed based on data from two different periods. The network has 122 nodes, and each node represents a street block (Figure 5). Figure 5 shows the difference in the network structure between weekdays and weekends, where the flow strength is represented by the color and thickness of the line of the edge: the wider the red line, the larger the flow strength, and a blue line indicates weak flow strength. Table 4 shows the ten travel activities with the highest flow volume on weekdays (19–21 May) and weekends. According to the information in the figure and table, we can summarize the following characteristics of the travel network:

From the network structure of a single weekday (Figure 5a,b) or the network structure of a period of several weekdays (Figure 5c), we can see that most travel activities occur between adjacent streets, which verifies that bike sharing is usually a short-distance travel method. The flow volume is relatively concentrated. In particular, the Huaxiang District Office and Xincun Sub-district Office tend to generate the greatest traffic flow during weekdays (Figure 5). The flow volume is relatively high in the area closed to Shibalidian District Office, Wangjing Development Street, Zuojiazhuang Sub-district Office, Panjiayuan Sub-district Office, Hujialou Sub-district Office and Liulitun Sub-district Office. These street blocks play an important role in the network structure on weekdays. The reason for the formation of the above network structure may be that on weekdays, people's travel purpose is often related to work, and shared bicycles are used as a method of short-distance commuting to avoid congested roads during peak hours. Therefore, destination streets with higher traffic may have more businesses or employment units, such as Wangjing Development Street, Huaxiang District Office, Gaobeidian District Office, etc.

On weekends, the network structure is relatively complex, and the events of travel flow referring to several street blocks are significantly greater than on weekdays. This shows that more people choose to use shared bicycles for long-distance travel activities on weekends. Meanwhile, the travel destinations have a stronger diversity. The flow on weekends is more scattered than that on weekdays, as can be seen from Table 4. Additionally, there are more node pairs, which can verify that the interaction between the street blocks on weekends is stronger.

*3.2. Characteristics Analysis of Travel Network*

The network density and degree of node accessibility of the spatial interactions during weekends and weekdays are computed in this paper. The in-degree and out-degree of the same street block are added, in order to calculate the node strength of all street blocks. Then, node strength is arranged in descending order. The importance of every node (street block) can be analyzed with the spatial interaction. The following conclusions can be drawn:

With the value of network density, the daily travel volume on weekdays tends toward stable. The network density on weekends is greater than that on weekdays (19–21 May). On weekends, the interaction between the various streets is more active, and the flow is also greater (see Table 5).

The differences in the travel network attributes are analyzed, as shown in Figure 6, where the X-axis shows the street IDs. From Figure 6a,b, the Y-axis shows the value of the node's out-degree or in-degree. From Figure 6c,d, the Y-axis represents the total degree value corresponding to the street. In Figure 6e, the Y-axis shows the value of the node's NFR.

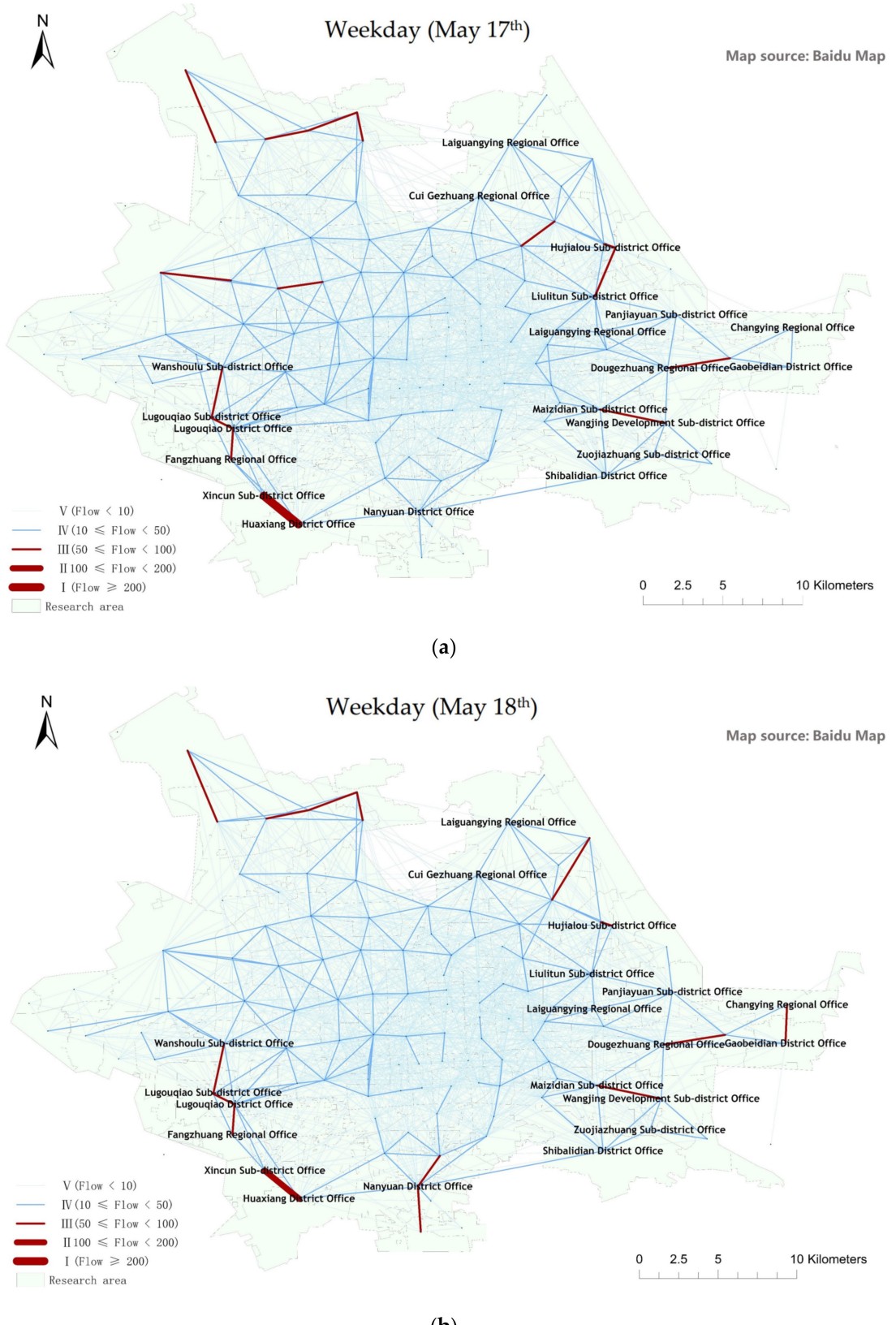

**Figure 5.** *Cont.*

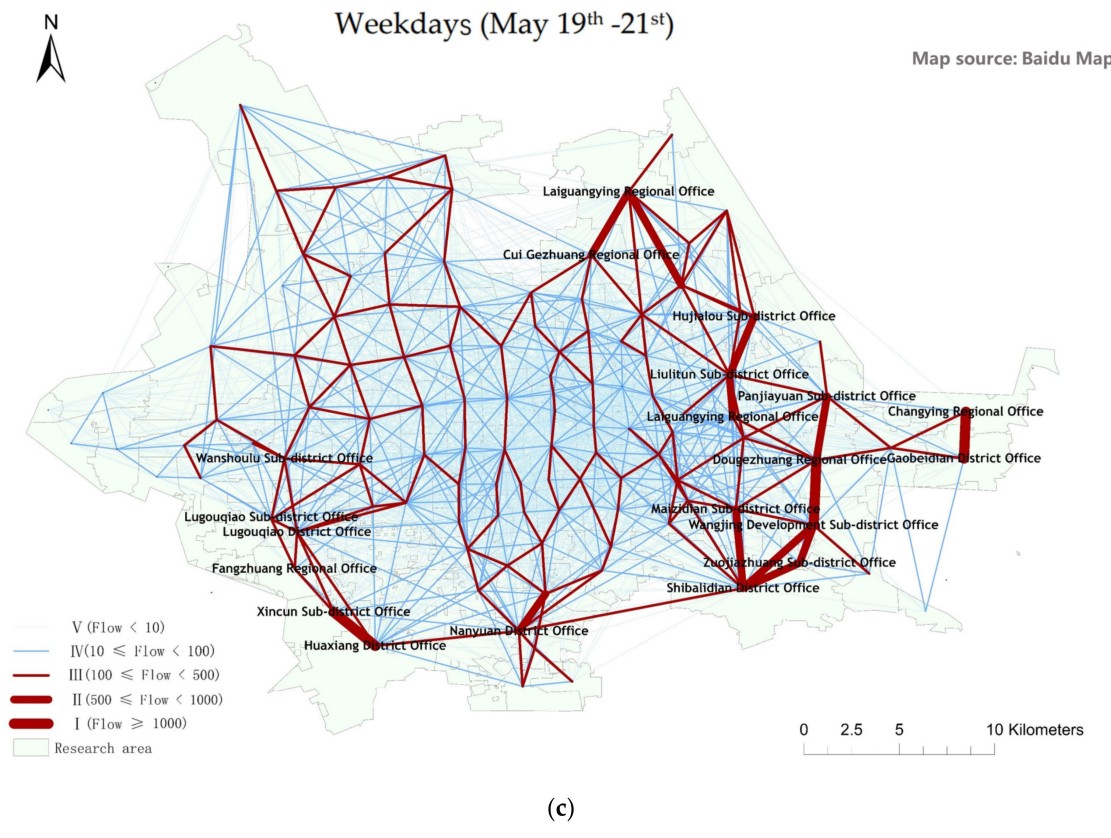

(**c**)

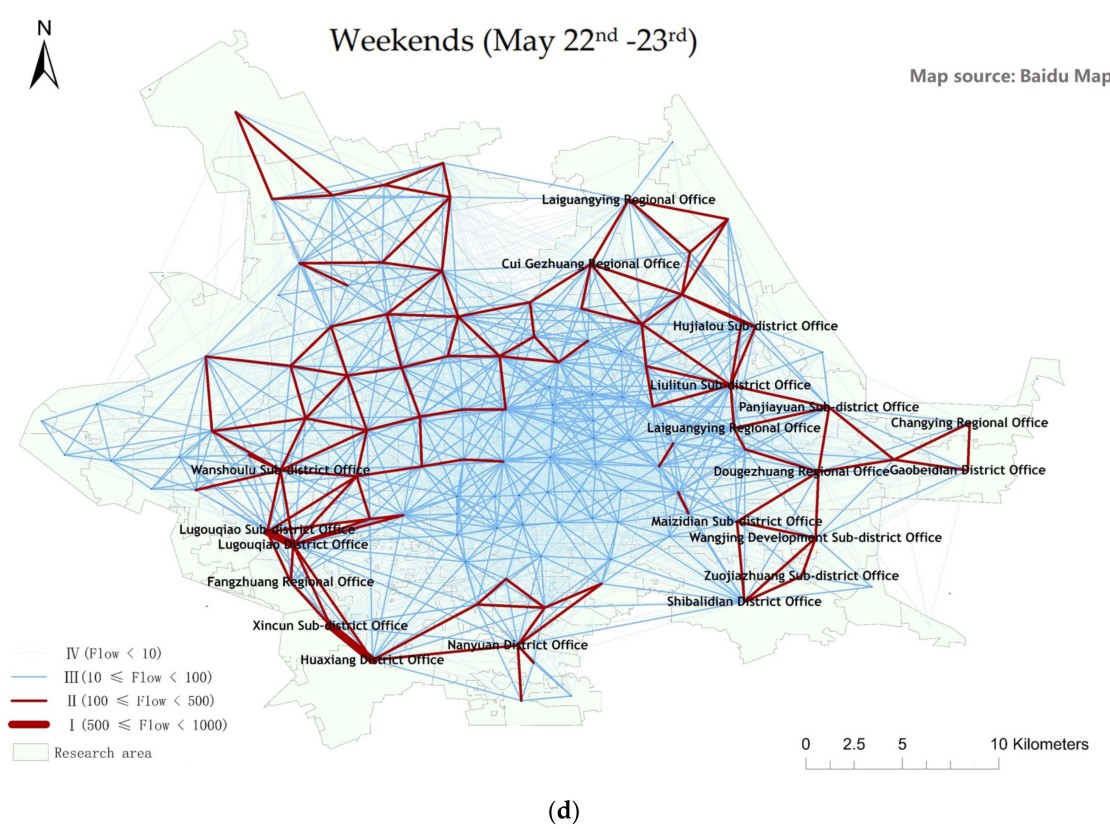

(**d**)

**Figure 5.** Bike travel network structure: (**a**) Weekday (17 May); (**b**) Weekday (18 May); (**c**) Weekdays (19–21 May); (**d**) Weekends (22–23 May).

**Table 4.** Travel volume ranking.

| Periods | Origin | Direction | Destination | Flow |
|---|---|:---:|---|---|
| Weekday (19–21 May) | Dougezhuang District Office | → | Wangjing Development Street | 1532 |
| | Xincun Sub-district Office | → | Huaxiang District Office | 1184 |
| | Changying District Office | → | Gaobeidian District Office | 1101 |
| | Panjiayuan Sub-district Office | → | Dougezhuang District Office | 891 |
| | Maizidian Sub-district Office | → | Shibalidian District Office | 853 |
| | Hujialou Sub-district Office | → | Liulitun Sub-district Office | 812 |
| | Laiguangying District Office | → | Cui Gezhuang District Office | 670 |
| | Wangjing Development Street | → | Shibalidian District Office | 645 |
| | Zuojiazhuang Sub-district Office | → | Shibalidian District Office | 612 |
| | Wangjing Development Street | → | Zuojiazhuang Sub-district Office | 604 |
| | Xincun Sub-district Office | → | Huaxiang District Office | 946 |
| Weekend (22–23 May) | Huaxiang District Office | → | Xincun Sub-district Office | 880 |
| | Lugouqiao District Office | → | Lugouqiao Sub-district Office | 565 |
| | Lugouqiao Sub-district Office | → | Lugouqiao District Office | 536 |
| | Jinzhan District Office | → | Hujialou Sub-district Office | 444 |
| | Hujialou Sub-district Office | → | Jinzhan District Office | 416 |
| | Qinghe Sub-district Office | → | Xisanqi Sub-district Office | 386 |
| | Shangdi Sub-district Office | → | Qinghe Sub-district Office | 371 |
| | Xisanqi Sub-district Office | → | Qinghe Sub-district Office | 367 |
| | Dongsheng District Office | → | Xisanqi Sub-district Office | 361 |

**Table 5.** Basic index statistics of the network.

| Periods | Number of Nodes | Average Node Strength | Network Density |
|---|---|---|---|
| Weekday (17 May) | 122 | 215.15 | 0.1660344 |
| Weekday (18 May) | 122 | 210.97 | 0.1608183 |
| Weekday (19–21 May) | 122 | 1275.68 | 0.2518629 |
| Weekend (22–23 May) | 122 | 1459.23 | 0.3504945 |

The network structure indicators of weekdays (19–21 May) and weekends are analyzed. From Figure 6, we can conclude that combined with the value of node strength, the in-degree and out-degree have a significant difference on weekdays. Meanwhile, at weekends, the in-degree and out-degree in the same street are close to each other. This verifies the phenomenon that residents have different travel purposes on weekdays and weekends. The purpose of residents' travels during weekdays is more clear, which results in larger differences in in-degree and out-degree values in some streets (Figure 6a,b). Regardless of the temporal scale, only a few streets have strong node strengths, and most streets have node strengths near the average. The node intensity of most streets on weekdays is between 500 and 2000, and the range of node strength between 500 and 2500 on weekends covers most of the streets. The degree of distribution of all streets is consistent with the power–law distribution, that is, it satisfies the scale-free characteristics of the complex network (Figure 6c,d).

Finally, according to our results, the value of NFR for most streets on weekdays is much larger than on weekends. On weekdays, the NFR of many streets shows large positive and negative values. On the one hand, these streets are hot spots in the network during weekdays, and many people choose to use these streets as the starting point or end of their trip. On the other hand, on weekdays, residents have a clearer potential willingness of travel from streets with negative NFR to streets with positive NFR. The larger the absolute value of the NFR of the street, the more likely the street has a single attribute between employment or residence, that is, the street is an employment area or a residential area. On weekends, the popularity of each street is small, and the NFR always fluctuates around the 0 mark, because the inflow and outflow of each street block are roughly equal. Additionally,

it shows that over the weekend, all street blocks interact with each other, and the volume of travel flow is also larger (see Figure 6).

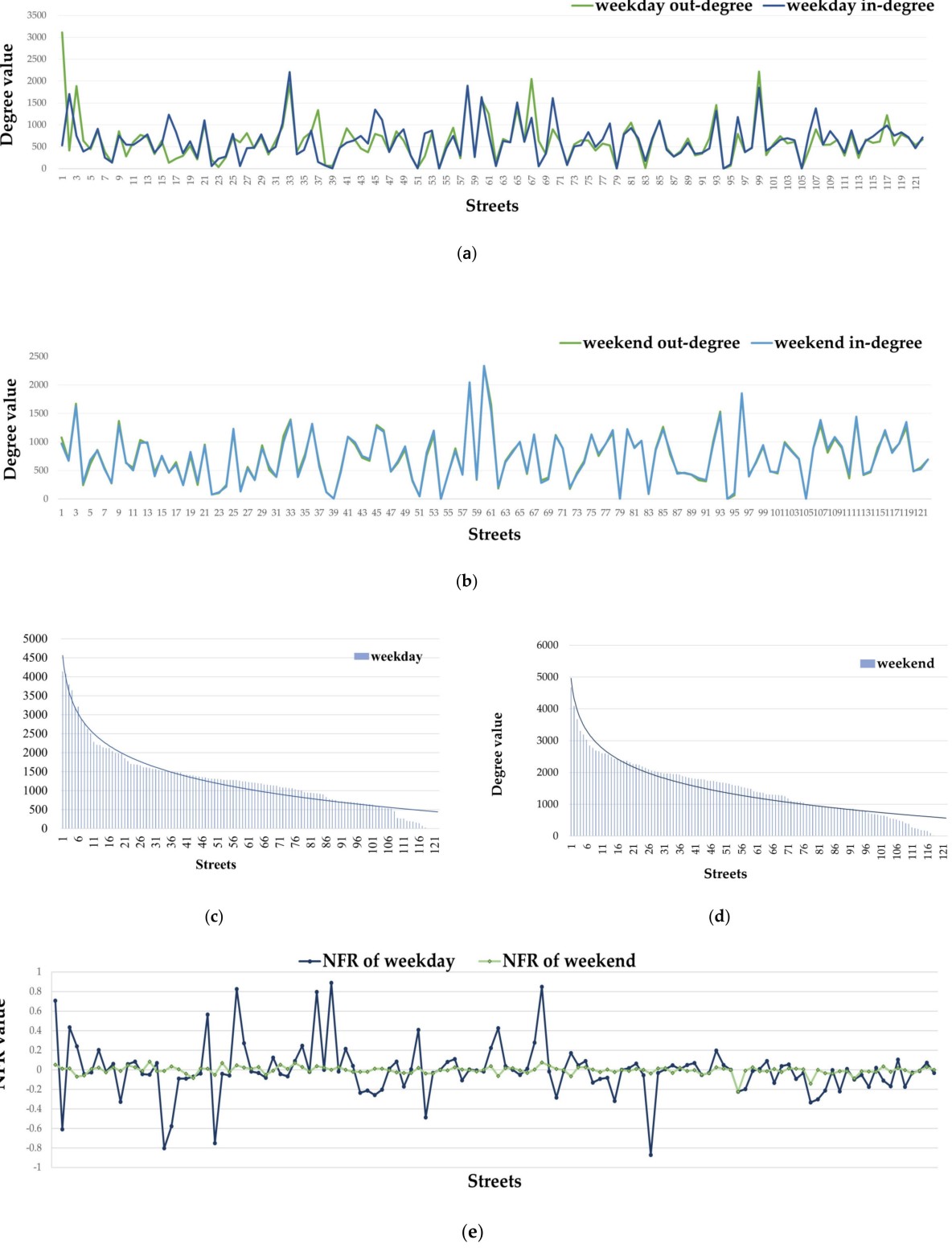

**Figure 6.** Analysis of travel network characteristics: (**a**) Analysis of in-degree and out-degree during weekday (19–21 May); (**b**) Analysis of in-degree and out-degree during weekend (22–23 May); (**c**) Analysis of node strength during weekday (19–21 May); (**d**) Analysis of node strength during weekend (22–23 May); (**e**) NFR (19–23 May).

### 3.3. Analysis of Influencing Factors of Travel Network

The *statnet* package in the R language is used to estimate, simulate, compare and test the ERGM model. The model can be downloaded from https://github.com/statnet/ergm (accessed on 20 May 2021). Additionally, the method of the classic Markov chain Monte Carlo maximum likelihood estimation (MCMC MLE) is used to train the model parameters. The evaluation indicators of the fitting effect adopt the Akaike Information Criterion (AIC) and Bayes Information Criterion (BIC). The smaller the corresponding index value, the better the model effect. The ERGM estimation results are shown in Table 6. The numerical value in Table 6 represents the result coefficient, the standard error corresponding to the value is in brackets, and, finally, P is the parameter for judging the hypothesis test result.

**Table 6.** Analysis of the influencing factors of the travel network.

| | Dependent Variable | | | |
|---|---|---|---|---|
| | Weekday | | | Weekend |
| | **17 May** | **18 May** | **19–21 May** | **22–23 May** |
| Network structure | | | | |
| Edges | −5.001 *** | −4.941 *** | −4.689 *** | −5.389 *** |
| | (0.0087) | (0.0090) | (0.0087) | (0.013) |
| Mutual | 2.66 *** | 2.801 *** | 2.305 *** | 4.37 *** |
| | (0.0041) | (0.0041) | (0.0036) | (0.0053) |
| Node attribute | | | | |
| Price | 0.00066 *** | 0.00062 *** | 0.00067 *** | 0.00067 *** |
| | (0.000062) | (0.000061) | (0.000054) | (0.000052) |
| Bden | 0.001003 *** | 0.00083 ** | 0.0016 *** | 0.0015 *** |
| | (0.00028) | (0.00027) | (0.00024) | (0.00021) |
| Rden | 0.000089 *** | 0.000096 *** | 0.00008 *** | 0.00011 *** |
| | (0.000018) | (0.000020) | (0.000017) | (0.000015) |
| Tra | 0.00025 *** | 0.00016 * | 0.00022 *** | 0.00017 *** |
| | (0.000065) | (0.000065) | (0.00006) | (0.000049) |
| SaD | 0.00015 *** | 0.00015 *** | 0.00016 *** | 0.00016 *** |
| | (0.000020) | (0.000021) | (0.000023) | (0.000015) |
| Fin | 0.00049 *** | 0.00036 ** | 0.0004 *** | 0.00025 ** |
| | (0.00011) | (0.00012) | (0.000098) | (0.000097) |
| Res | 0.00048 *** | 0.00052 *** | 0.00057 *** | 0.00049 *** |
| | (0.00012) | (0.00011) | (0.000099) | (0.00009) |
| Edu | 0.00012 *** | 0.00014 ** | 0.00013 * | 0.00018 ** |
| | (0.000089) | (0.000079) | (0.00007) | (0.000062) |
| AIC | 10575 | 10767 | 13838 | 12334 |
| BIC | 10651 | 10843 | 13914 | 12410 |

Note: * $p < 0.1$, ** $p < 0.05$, *** $p < 0.01$.

From the fitting results of pure network structure variables, it can be seen that the coefficients of the edge statistical items are all negative values, which means the network density is less than 50%. Existing research generally regards this item as a constant item without explanation [43]. The variable Mutual can reflect the possibility of interaction between pairs of street blocks. The larger the value, the more interaction between these two street blocks. Meanwhile, the value of the variable Mutual during weekends is higher than on weekdays (19–21 May), which shows that the interaction between the streets on weekends is stronger (Table 6). The results are also consistent with the results obtained from the network structure analysis.

From the fitting results of the node attribute variables in Table 6, it can be seen that all variables have obvious positive coefficients, which indicates that the influencing factors selected in this paper all have a positive impact on the formation of the shared bicycle travel network. The coefficient of Bden is significantly higher than the coefficients of other influencing factors, indicating that building density has a great impact on promoting travel behavior, whether on weekdays or weekends. Additionally, with the accumulation of

weekdays, the degree of influence of building density (Bden) becomes more and more obvious (Figure 7a). The larger the density of buildings in the street, the higher the demand for shared bicycles. The coefficients of Price and Res are almost equal, showing more importance than the other five influencing factors, except Bden. Factors such as Eco, Rden, Tra, SaC and Edu all have a positive correlation, but the impact is relatively low (Figure 7a,b).

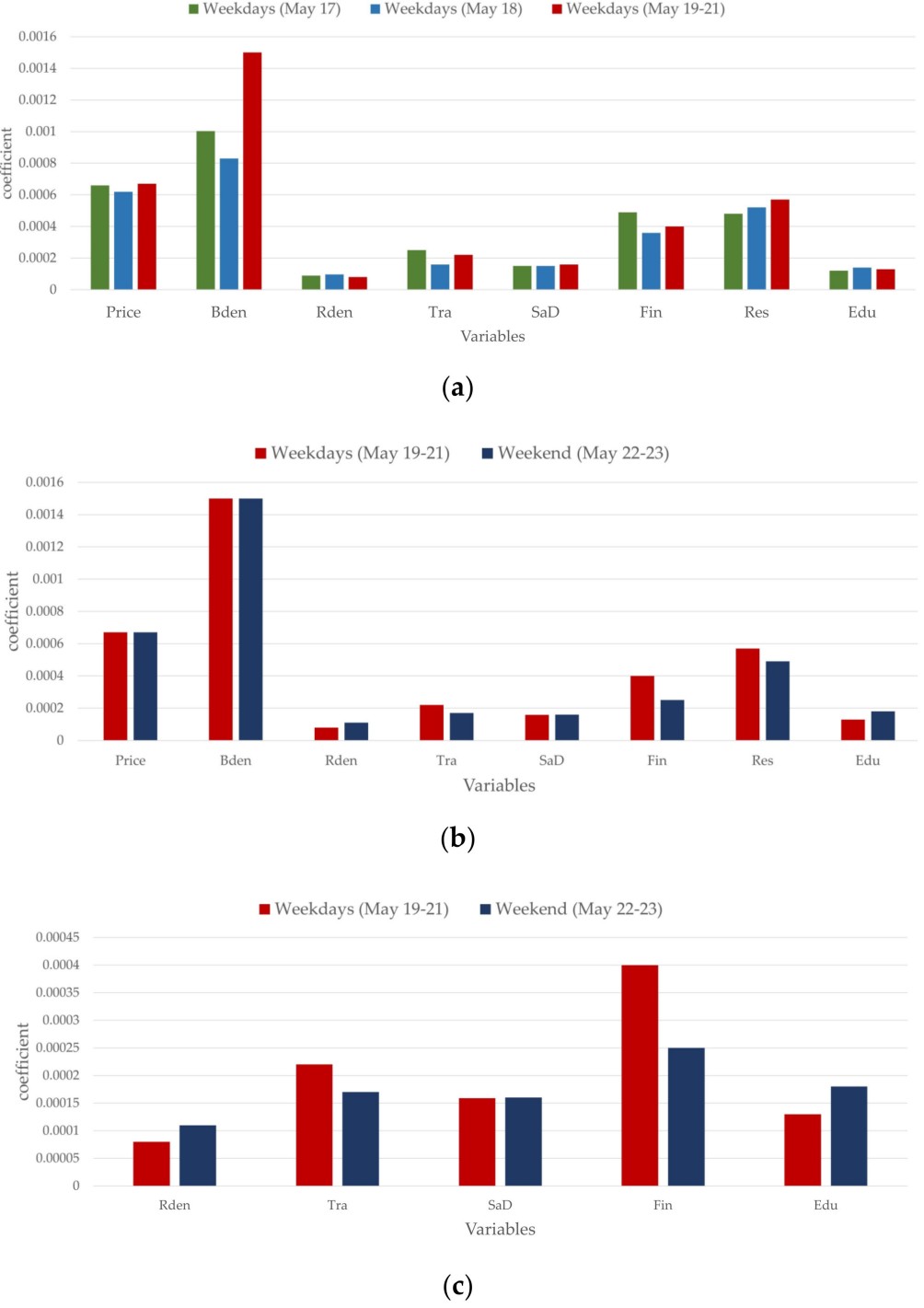

(a)

(b)

(c)

**Figure 7.** ERGM estimation results: (**a**) Coefficient values of all the influencing factors on weekdays; (**b**) Comparison of coefficient values of all influencing factors on weekdays and weekends; (**c**) Coefficient values of influencing factors other than Price, Bden and Res.

As an important indicator to measure the level of urban development, building density determines the comprehensive development level of streets to a certain extent. The higher the development of the street, the more frequent the movement of people. There is a clear correlation between house prices and travel, which is consistent with Lu J's conclusion [56]. In addition, considering the special rental housing situation in Beijing, in which the renting population accounts for one-third of the population of permanent residents, such people tend to have a greater demand for shared bicycles, which increases the traffic on the streets. There is a certain positive correlation between the number of residential areas and the size of the resident population, resulting in a high positive coefficient of Res.

This paper compares and analyzes the influencing factors of working days (19th–21st) and weekends (Figure 7b,c). From the aspects of temporal analysis, comparing the coefficients of various factors on weekdays and weekends, it can be concluded that the coefficients of Bden, Price, and Res reach the same peak in the two periods, indicating that these factors are effective for people traveling between streets in either part of the week (Figure 7b). The remaining five coefficients are Eco, Rden, Tra, SaC and Edu. From the weekday results, it can be seen that the coefficient of the financial service factor is larger than the coefficients of the other four factors (Figure 7c). Financial service facilities are used to measure the financial service capacity of the street, which can reflect the economic development level of the street to a certain extent, and are the most important factor to distinguish between weekends and weekdays. The more convenient the financial services of the street, the more shared bicycle flow can be generated on weekdays. From the weekend estimation results, it can be seen that the coefficients of the above five influencing factors are almost the same. The formation of the shared bicycle travel network is the result of a combination of many factors. A person's choice to use a shared bicycles to reach their destination is always made according to their own needs.

## 4. Discussion

Based on the theory of complex networks, a spatial interaction network is constructed during weekdays and weekends, and the results are analyzed in terms of three aspects: network structure, network statistical indicators, and influencing factors.

(i) Through the analysis of the network structure, it is found that on weekdays, the spatial pattern of the network is relatively stable, most of the flow is generated between adjacent blocks, and the flow is relatively concentrated at several specific places. On weekends, the network pattern is relatively complicated, with more flow across street blocks. To be specific, on weekdays, shared bicycles are usually the choice for short-distance travel, and the purpose of residents' travel is stronger. On weekends, more people may choose to share bicycles for longer-distance travel, the starting and ending points of travel are diverse, and the inter-street interaction is stronger. (ii) The network density and degree of node accessibility of the spatial interaction network during weekdays and weekends are computed in this paper. It is found that the network density on weekends is greater than that on weekdays. The number of accessible degrees in the same street block varies greatly on weekdays, but remains almost the same at weekends. The distribution of node strength presents a power–law distribution on both weekdays and weekends, which is consistent with the scale-free characteristics of complex networks. NFR calculation results show that residents travel more purposefully during the weekdays, leading to a sharp increase or decrease in the popularity of certain neighborhoods. The magnitude of the absolute value of the NFR also determines whether the street is an employment area or a residential area. On weekends, residents' travel patterns are relatively complicated. Since the inflow and outflow of each street block are roughly equal, and the increase or decrease in popularity is small. (iii) According to the results of ERGM, the reciprocity (Mutual) on weekends is much higher than that on weekdays, which verified the results suggesting that the interactivity between streets on weekends is stronger. The influencing factors selected in this paper all have a positive impact on the formation of the network. Among them, the influence coefficient of building density (Bden) is the largest. Additionally, with

the accumulation of weekdays, the influence of building density (Bden) becomes more obvious. It can be seen that the building density of the street determines people's choice of shared bicycles as a mode of travel. The effects of housing prices (Price) and the number of residential areas (Res) are the second in importance only to building density (Bden). By comparing the estimation results on weekdays and weekends, it can be seen that on weekdays, street blocks with more convenient financial services are more likely to generate travel and spatial interaction flow. On weekends, financial services (Fin), road network density (Rden), transportation (Tra), shopping and dining (SaD) and education institution (Edu) have roughly the same impact.

Based on the above research results, this paper can provide corresponding decision-making assistance for street management departments and bicycle-sharing companies, and provide a reference for urban traffic management and social management to formulate time-sharing management policies and measures. First, bicycle-sharing companies can schedule vehicles in advance according to the increase or decrease in the heat of a street to meet the balance of supply and demand. Streets with high heat should focus on improving the BSS and adjust relevant policies in a timely manner to prevent traffic problems caused by excessive crowds. Second, priority should be given to deploying more bike-sharing stations in areas with higher building density. In addition, on weekdays, priority should be given to setting up more shared bicycle gathering points near financial institutions. During holidays, appropriately increasing the number of bicycles that can be used near transportation, shopping, and catering locations, financial services, scientific research and education institutions, etc., can effectively increase the utilization rate of shared bicycles and enable residents to travel. This study not only enriches the methodology of the research on the influencing factors of shared bicycles, but also regards the influencing factors as an important feature to distinguish weekdays from weekends. While validating previous research results, there are many other new discoveries. Nonetheless, our research has potential for improvement in influencing factor selection, travel data attributes, and network construction.

First, it is possible to continue to expand the influencing factors that can be studied. The variables related to urban areas are still being refined. Future research should also include sports facilities, municipal media, road congestion, weather, the COVID-19 pandemic, and more. Second, the data should be further refined in the future to increase the applicability of the conclusions. It is necessary to increase the resident attributes of the travel data, such as the social attributes (occupation, gender, and age) of residents for accurately analyze of the travel purposes. Third, we can continue to refine how complex networks are built. The network constructed with streets as nodes cannot directly study a specific shared bicycle parking spot in the area. In the future, such hot spots can be used as network nodes to make the results more specific. In addition, it is necessary to continuously strengthen cooperation and interaction with other disciplines, so that the research results have greater social and economic value and significance. Finally, the shared bike data in this article covers a full week, while in order to found more valuable rules of the shared bikes travel, we will expand the time coverage in the next step. In addition, most of the data used in this paper is confidential data, so the reproducibility of this paper is limited. In the future study, we will use public datasets to improve the reproducibility of the paper's experiments.

**Author Contributions:** Conceptualization, Lujin Hu and Zheng Wen; Data curation, Zheng Wen and Jing Hu; Methodology, Lujin Hu; Project administration, Jian Wang; Validation, Zheng Wen; Visualization, Zheng Wen; Writing—original draft, Zheng Wen; Writing—review & editing, Lujin Hu, Jian Wang and Jing Hu All authors have read and agreed to the published version of the manuscript.

**Funding:** This work was supported by the National Key Research and Development Program of China (Grant No. 2020YFD1100201).

**Institutional Review Board Statement:** Not applicable.

**Informed Consent Statement:** Not applicable.

**Data Availability Statement:** Not applicable.

**Conflicts of Interest:** The authors declare no conflict of interest.

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
