# Peer review of "Spatial Interaction Analysis of Shared Bicycles Mobility Regularity and Determinants: A Case Study of Six Main Districts, Beijing"

_ijgi, doi:10.3390/ijgi11090477_

Round 1

Reviewer 1 Report

Please see the attached report for comments and suggestions.

Reviewer 2 Report

In the initial part, the novelty of the research should be emphasised, while in the concluding part, the limitations and future research steps should be underlined.

All acronyms should be included in extended form when they are mentioned for the first time in the text.

It is considered appropriate to

1) reword the title to make it more appealing 

2) insert figures 2 and 3 

3) include more bibliographical references about the evolution of BSSs and the factors analysed in the literature to better describe the demand for transport.We suggest reading the following research works

a) Torrisi, V., Ignaccolo, M., Inturri, G., Tesoriere, G., & Campisi, T. (2021). Exploring the factors affecting bike-sharing demand: Evidence from student perceptions, usage patterns and adoption barriers. Transportation Research Procedia52, 573-580.

b) Politis, I., Fyrogenis, I., Papadopoulos, E., Nikolaidou, A., & Verani, E. (2020). Shifting to shared wheels: Factors affecting dockless bike-sharing choice for short and long trips. Sustainability12(19), 8205.

c)Li, X., Zhang, Y., Du, M., & Yang, J. (2019). Social factors influencing the choice of bicycle: Difference analysis among private bike, public bike sharing and free-floating bike sharing in Kunming, China. KSCE Journal of Civil Engineering23(5), 2339-2348.

4) Check the numbering of the figures

5) Insert more commentary to accompany Figures 4 and 7. 

6)correct typos and grammatical errors in the text from the abstract 

Reviewer 3 Report

Thank you very much for the opportunity to review the article. The subject of the paper is interesting, but the article requires improvement.

It is worth emphasizing the novelty of the research and the authors' contribution to  theory and practice. What gap have the authors found in the analyzes so far? I suggest adding some hypotheses to your research.

The research results should be supported by the analysis of the cause and effect relationships. Moreover, the research results should be linked with the analyzes of the results obtained so far.

The list lacks detailed recommendations for the economy and environment, as well as future research directions. It is worth emphasizing the advantages of the research results.

Round 2

Reviewer 1 Report

Following the revision of the paper, its quality has improved significantly. However, there is still potential for improvement in some sections. 

Introduction

The first half of the introduction (~ line 94) is in significant need of revision. This mainly concerns the outline, the technical terms used, and the selection of related sources as well as some parts of the actual content.

Outline. There is a rapid change between statements regarding bike-sharing and travel behavior in general. For example, in lines 49 to 52 the authors write about research in travel behavior, the cited literature, however, is only related to bike-sharing systems. From line 54, they begin to discuss recent literature on bike-sharing influencing factors. However, the most relevant factors were already given in lines 49 to 52 (incorrectly related to general travel behavior). Thus the statements are repeated. In line 75 they suddenly switch to traffic flow in general again. Without bike-sharing reference, the statement seems out of place and does not contribute to the rest of the text. Subsequently, the statements refer again to bike-sharing systems. Lack of context to the rest of the text also applies to liness 45 to 48 (What has this to do with bike-sharing or the aim of the paper) and 63 to 64 (Context? Also lack of literature as already mentioned in the first review).

Terms. In lines 35 to 38 the authors write about problems of supply-demand matches during peak hours. From the rewier's point of view, this problem is related to the so-called and well-known bike-sharing rebalancing problem. There are dozens of related literature which could be cited rather than a single master's thesis.  In line 52, the authors state that "shared bicycles are a form of urban travel behavior". This is not correct as the bike is a mode of transport.

Additions. In lines 34 to 34, the authors state that "BSS is also equipped with [...] GPS [...], contributing much data for urban flow analysis in cities." This statement must be limited, as it is not universally valid. 

Materials and Methods

Although this section has been adapted in accordance with the notes, it still does not meet the criteria of transparent science. In their reply to the reviewer, the authors wrote that the data source for the bike-sharing data is subject to a confidentiality agreement. in addition, they provided sources for the POIs. However, neither of these can be found in the paper and is therefore not comprehensible to the actual reader. Furthermore, the use of publicly available data should be aspired in the sense of reproducibility. This point needs major revisions.

In line 183 to 185 it is stated that after pre-experiments the week from May 17th to May 23rd was chosen. However, it is not stated why this week was choosen and why this week is representative for the general utilization of the system. Further, it is questionable whether it is possible to make general statements about an entire system or its influencing factors on the basis of data from only a single week. The observation period should be significantly extended. If this cannot be done within the framework of the current paper, it is at least to be listed as a further limitation and outlook for future works.

In line 186 data collection intervals are missing. In line 187 it is stated that working days were divided to enable better analysis. However, it is not explained why the division allows a better analysis. The legend of figure 2 is not readable. Why is only the traffic flow from 17-20 o'clock witnessed in the figure? Figure 3 is also not readable.

Results/Discussion

In general, the limitations mentioned above should be included and discussed. Further, there are some additional things to revise.

In lines 440 and 529 the authors write about the "purpose" of residents' travels during the week. However, the response is not further mentioned or analysed. The traffic flows show comuting patters for weekdays. Perhaps this should be explored in more detail.

In line 580 it is stated that the "data should be further refined". However, this is not specified in any way. 

Reviewer 2 Report

the manuscript still has some grammatical errors 

It is necessary to include the sources of the maps used in the figures and to include these in high resolution with readable text

It is necessary to check the formatting of image captions 

Once this is corrected,the manuscript will be eligible for publication 

Round 3

Reviewer 1 Report

After the revision, the quality of the paper has improved significantly. However, there are still some points that require further revision.

Figure 1 on line 162: The dashed lines do not match the colored area. It looks as if they have "slipped".

In lines 179 the authors write about a complete continuous week and that this was one of the reasons to choose this week as observation period. However, it becomes not clear what they mean with continuous in this regard.

Figure 2 on line 197: The legends are still not readable.

Figure 3 on line 216: The legend is only hardly redable.

Figure 5 on line 402: The legends from a and b are different to c and d with regard to layout and scaling, respectively. To ensure comparability of the days, a consistent legend would be appropriate. Comparability is further complicated by the fact that a and b contain individual days and c and d contain the flows of several days, whereby it is not clear which flows were generated on which day.
